# Copper-Catalyzed One-Pot Synthesis of *N*-Sulfonyl Amidines from Sulfonyl Hydrazine, Terminal Alkynes and Sulfonyl Azides

**DOI:** 10.3390/molecules26123700

**Published:** 2021-06-17

**Authors:** Yu Zhao, Zitong Zhou, Man Chen, Weiguang Yang

**Affiliations:** The Marine Biomedical Research Institute, Guangdong Medical University, Zhanjiang 524023, China; gdmuzy@163.com (Y.Z.); zzt15766229745@163.com (Z.Z.); chenman66@126.com (M.C.)

**Keywords:** amidines, multicomponent reactions, CuAAC/ring-opening, *N*-sulfonylketenimines, nucleophilic addition

## Abstract

*N*-Sulfonyl amidines are developed from a Cu-catalyzed three-component reaction from sulfonyl hydrazines, terminal alkynes and sulfonyl azides in toluene at room temperature. Particularly, the intermediate *N*-sulfonylketenimines was generated via a CuAAC/ring-opening procedure and took a nucleophilic addition with the weak nucleophile sulfonyl hydrazines. In addition, the stability of the product was tested by a HNMR spectrometer.

## 1. Introduction

Amidine derivatives are important privileged scaffolds in medicinal chemistry [1,2,3], synthetic chemistry [4] and an important pharmacophore in drug discovery [5,6]. One subset of such compounds is *N*-sulfonyl amidine derivatives that show a prolific set of biological activities, including antifungal (I) [7], anticancer (II) [8], antiresorptive (III and IV) [9,10,11], antiproliferative (V) [12], dopamine transporter inhibitors (VI) [13] (Figure 1), etc. [14,15]. Therefore, the establishment of robust synthetic approaches for the preparation of *N*-sulfonyl amidines and their functionalizations is highly required.

Classical types of reactions have focused on the preparation of *N*-sulfonyl amidines involved in the reaction of cyclic thioamides and thioacetamide derivatives with sulfonyl azides [14,16,17,18], the phosphite-mediated Beckmann-like coupling of oximes and *p*-toluenesulfonyl azide [19], sulfonamide derivatives condensation with DMF–DMA [20], the sulfonamide reaction with formamide [21] and the sulfonyl ynamide rearrangement [22]. The most efficient method is the Cu-catalyzed multicomponent reaction of terminal alkynes, sulfonyl azides and amines, which has been applied to synthesize numerous oxygen-containing and nitrogen-containing heterocyclic compounds [23,24,25,26,27,28,29,30,31]. The ketenimine intermediate generated by Cu-catalyzed alkynes and sulfonyl azides [31,32,33] could take a nucleophilic addition reaction with most amines, as show in Scheme 1, including aliphatic primary amines [34,35,36], aliphatic secondary amines [37,38], aliphatic tertiary amines [39,40], quaternary amine salts [41], imines [27], nitrogenous heterocyclic compounds [42,43,44,45], urea derivatives [46], oximes [47], sulfoximines [48] and enyl amine [49,50]. However, to our knowledge, there are few previous works that used the weak nucleophile sulfonyl hydrazines for this method. Herein, the Cu-catalyzed one-pot synthesis of *N*-sulfonyl amidines from sulfonyl hydrazine, terminal alkynes and sulfonyl azides was reported.

## 2. Results

We began our investigation by examining the synthesis of 4-methyl-*N*-(2-phenyl-1-(2-(1-phenylethylidene)-1-tosylhydrazinyl) ethylidene)benzenesulfonamide **4a** via 4-methyl-*N*′-(1-phenylethylidene)benzenesulfonohydrazide **1a**, ethynylbenzene **2a** and *p*-tosyl azide **3a**. The reaction was carried out in the presence of CuI and Et_3_N in CH_2_Cl_2_ at room temperature for 1 h, and **4a** was isolated in a 78% yield (Table 1, entry 1). Based on this finding, the reaction conditions were screened. First, several catalysts were screened, and most Cu-catalysts exhibited a high catalytic reactivity in this reaction, whether Cu^I^-catalysts or Cu^II^-catalysts (Table 1, entries 2–6). Other catalysts such as AgTFA failed to produce the desired product (Table 1, entries 7). Then, the effects of different bases were evaluated, and the screening results revealed that the use of Et_3_N achieved a superior result compared to DMAP, DIPEA, pyridine and the other bases (Table 1, entries 8–12). Finally, the solvents were screened, and a lower or comparable yield was obtained when CHCl_3_, DCE, MeCN, THF, DMSO and DMF were used as solvents, while toluene gave **4a** the highest yield of 84% (Table 1, entry 13–19). Encouraged by this promising result, we tracked the reaction by TLC and found that it could be completed in less than an hour at room temperature (Table 1, entry 20–23).

With the optimized reaction conditions obtained, the substrate diversity with the sulfonyl hydrazines **1** was tested first. As shown in Scheme 2, the R^1^ electron effects of the substituents **1** had slight influences. For example, substrates bearing 4-OMe-C_6_H_4_, 4–Me-C_6_H_4_, 2-naphthyl and 2-tetra-hydronaphthalyl were examined, and the 72–88% yields of **4a**–**4e** were isolated. The R^2^ of substrates **1** bearing the 2,4,6-trimethylphenyl group also can obtain **4f** in a good yield of 80%. However, when changing the substrates **1** to other sulfonyl hydrazines, such as **1g**–**1k**, it could not obtain the desired products and give decomposed or complex compounds. Next, the scopes and limitations of terminal alkynes **2** and sulfonyl azides **3** were examined. An aryl-substituted, aliphatic or 2-thienyl terminal alkynes and aryl-substituted or aliphatic sulfonyl azides can smoothly obtain the corresponding products **4g**–**4m** with yields of 73–89% and **4n**–**4q** with yields of 78–86%, in which both the substituents led to high yields and were influenced slightly.

The structure of **4a** was confirmed by X-ray crystallography (Figure 2, CCDC deposition number 2075031).

Curiously, we found that the separated products in the solvent were unstable and would decompose. Thus, the stability of product **4a** was tested by a HNMR spectrometer. As shown in Figure 3, the products dissolved in DMSO were relatively stable in the first four days, and the decomposition complex could be observed starting from the fifth day; then, the concentration of byproducts became thicker day by day. After a month, the system was relatively stable, and the decomposition was slow. Therefore, it is recommended that products **4a**–**4q** should be dried and stored at a low temperature.

## 3. Experimental

### 3.1. General Information

All melting points were determined on a Yanaco melting point apparatus and were uncorrected. IR spectra were recorded as KBr pellets on a Nicolet FT-IR 5DX spectrometer. All spectra of ^1^H NMR (400 MHz) and ^13^C NMR (100 MHz) were measured on a 400 MHz Bruker spectrometer using DMSO-*d_6_* or CDCl_3_ as the solvent, with tetramethylsilane (TMS) as the internal standard, at room temperature. Chemical shifts are given in δ relative to TMS, and the coupling constants *J* are given in Hz. HRMS were obtained on a Bruker micrOTOF-Q II spectrometer. All commercially available reagents were purchased from Sigma-Aldrich, Acros, Aladdin, TCI, Alfa, Innochem in China and were used without further purification. All reactions were carried out in dried reaction tube (25 mL). The original ^1^H and ^13^C NMR spectra are available in Appendix A.

### 3.2. Compound Characterizations and Preparations

4-methyl-*N*-((*E*)-2-phenyl-1-(2-((*E*)-1-phenylethylidene)-1-osylhydrazineyl) ethylidene) benzenesulfonamide (**4a)**. 4-methyl-*N*′-(1-phenylethylidene) benzenesulfonohydrazide (**1a**) (0.114 mg, 0.50 mmol) was mixed with CuI (9.5 mg, 0.05 mmol) in 1-mL toluene. Then, ethynylbenzene (**2a**) (76.5 mg, 0.75 mmol), TsN_3_ (147.8 mg, 0.75 mmol) and TEA (101 mg, 1.0 mmol) were mixed in toluene (2 mL). After stirring at room temperature for 1 h and concentrated under reduced pressure, the mix was purified a flash chromatography (petroleum ether/ethyl acetate: 7:1) to give product **4****a** as a white solid, mp 143–144 °C. IR (KBr) *ν* 3063, 1564, 1492, 1442, 1309, 1145, 1082 cm^−1^; ^1^H NMR (400 MHz, DMSO-*d_6_*) *δ* 7.82 (d, *J* = 8.0 Hz, 2H), 7.62 (t, *J* = 8.0 Hz, 3H), 7.53–7.46 (m, 6H), 7.28–7.21 (m, 5H), 7.01 (d, *J* = 6.8 Hz, 2H), 4.14 (s, 2H), 2.48 (s, 3H), 2.42 (s, 3H), 1.73 (s, 3H); ^13^C NMR (100 MHz, DMSO-*d_6_*) *δ* 182.7, 165.2, 145.6, 143.6, 138.6, 135.0, 134.0, 133.1, 132.4, 129.7 (2C), 129.6 (2C), 128.9 (2C), 128.8 (2C), 128.6, 128.5 (2C), 127.8 (2C), 127.2, 126.5 (3C), 21.3 (3C), 17.7; HRMS (ESI-TOF) (*m*/*z*). Calcd for C_30_H_29_N_3_O_4_S_2_, [M + H]^+^ 560.1672; found 560.1675.

The products **4b**–**4q** were prepared by a similar procedure.

4-methyl-*N*-((*E*)-2-phenyl-1-(2-((*E*)-1-(*p*-tolyl)ethylidene)-1-tosylhydrazineyl) ethylidene)benzenesulfonamide (**4b**). White solid, mp 153–155 °C. IR (KBr) *ν* 3062, 1594, 1568, 1307, 1172, 1147, 1084 cm^−1^; ^1^H NMR (400 MHz, DMSO-*d_6_*) *δ* 7.72 (d, *J* = 8.0 Hz, 2H), 7.62 (d, *J* = 8.0 Hz, 2H), 7.47 (t, *J* = 7.8 Hz, 4H), 7.31 (d, *J* = 8.0 Hz, 2H), 7.27 (d, *J* = 8.0 Hz, 2H), 7.24–7.19 (m, 3H), 7.00 (d, *J* = 6.8 Hz, 2H), 4.19 (s, 2H), 2.47 (s, 3H), 2.42 (s, 3H), 2.39 (s, 3H), 1.69 (s, 3H); ^13^C NMR (100 MHz, DMSO-*d_6_*) *δ* 182.3, 165.3, 145.5, 143.5, 142.6, 138.6, 134.0, 133.1, 132.3, 129.7 (2C), 129.6 (2C), 129.3 (2C), 128.9 (2C), 128.6 (2C), 128.5 (2C), 127.8 (2C), 127.1, 126.5 (3C), 21.2 (3C), 17.7; HRMS (ESI-TOF) (*m*/*z*). Calcd for C_31_H_31_N_3_O_4_S_2_, [M + H]^+^ 574.1829; found 574.1831.

*N*-((*E*)-1-(2-((*E*)-1-(4-methoxyphenyl)ethylidene)-1-tosylhydrazineyl)-2-phenylethylidene)-4-methylbenzenesulfonamide (**4c**). White solid, mp 141–143 °C. IR (KBr) *ν* 3063, 1590, 1494, 1289, 1173, 1141, 1085 cm^−1^; ^1^H NMR (400 MHz, DMSO-*d_6_*) *δ* 7.81 (d, *J* = 8.8 Hz, 2H), 7.62 (d, *J* = 8.0 Hz, 2H), 7.47 (t, *J* = 7.8 Hz, 4H), 7.27 (d, *J* = 8.0 Hz, 2H), 7.23–7.18 (m, 3H), 7.05–6.99 (m, 4H), 4.49 (s, 2H), 3.85 (s, 3H), 2.47 (s, 3H), 2.42 (s, 3H), 1.67 (s, 3H); ^13^C NMR (100 MHz, DMSO-*d_6_*) *δ* 181.5, 165.3, 162.6, 145.5, 143.5, 138.7, 134.0, 133.1, 132.3, 129.7 (2C), 129.6 (2C), 129.5 (2C), 128.9 (2C), 128.6, 128.5 (2C), 127.3, 127.1, 126.5 (3C), 114.1, 55.6, 21.2 (3C), 17.2; HRMS (ESI-TOF) (*m*/*z*). Calcd for C_31_H_31_N_3_O_5_S_2_, [M + H]^+^ 590.1778; found 590.1782.

4-methyl-*N*-((*E*)-1-(2-((*E*)-1-(naphthalen-2-yl)ethylidene)-1-tosylhydrazineyl)-2-phenylethylidene)benzenesulfonamide (**4d****)**. White solid, mp 172–173 °C. IR (KBr) *ν* 3056, 1590, 1574, 1494, 1359, 1305, 1144, 1084 cm^−1^; ^1^H NMR (400 MHz, DMSO-*d_6_*) *δ* 8.39 (s, 1H), 8.07 (d, *J* = 8.0 Hz, 1H), 8.02 (t, *J* = 7.2 Hz, 3H), 7.68–7.61 (m, 4H), 7.53–7.46 (m, 4H), 7.29 (d, *J* = 8.0 Hz, 2H), 7.25–7.17 (m, 3H), 7.02 (d, = 7.2, 2H), 4.34 (s, 2H), 2.48 (s, 3H), 2.43 (s, 3H), 1.87 (s, 3H); ^13^C NMR (100 MHz, DMSO-*d_6_*) *δ* 182.3, 165.3, 145.6, 143.6, 138.6, 134.7, 134.0, 133.1, 132.4 (2C), 129.6 (2C), 129.4 (2C), 129.3 (2C), 128.9 (2C), 128.6, 128.5 (2C), 128.3, 128.2, 127.7, 127.2, 127.0, 126.5 (3C), 123.7, 21.2 (3C), 17.6; HRMS (ESI-TOF) (*m*/*z*). Calcd for C_34_H_31_N_3_O_4_S_2_, [M + H]^+^ 610.1829; found 610.1832.

4-methyl-*N*-((*E*)-2-phenyl-1-(2-((*E*)-1-(5,6,7,8-tetrahydronaphthalen-2-yl)ethylidene)-1-tosylhydrazineyl)ethylidene)benzenesulfonamide (4e). White solid, mp 173–174 °C. IR (KBr) *ν* 3062, 3030, 1590, 1494, 1370, 1176, 1145, 1083 cm^−1^; ^1^H NMR (400 MHz, DMSO-*d_6_*) *δ* 7.61 (d, *J* = 8.0 Hz, 2H), 7.50 (d, *J* = 8.0 Hz, 1H), 7.46 (d, *J* = 6.4 Hz, 5H), 7.28–7.16 (m, 6H), 6.99 (d, *J* = 8.0, 2H), 4.02 (s, 2H), 2.78 (s, 4H), 2.47 (s, 3H), 2.42 (s, 3H), 1.76 (s, 4H), 1.69 (s, 3H); ^13^C NMR (100 MHz, DMSO-*d_6_*) *δ* 182.6, 165.3, 145.6, 143.6, 141.8, 138.6, 137.2, 134.0, 133.1, 132.4, 129.7 (2C), 129.6 (2C), 129.3, 128.9 (2C), 128.7 (2C), 128.6 (2C), 128.4, 127.2, 126.5 (2C), 124.9, 28.9 (2C), 22.6, 22.5, 21.3, 21.2, 17.6 (2C); HRMS (ESI-TOF) (*m*/*z*). Calcd for C_34_H_35_N_3_O_4_S_2_, [M + H]^+^ 614.2142; found 614.2145.

*N*-(1-(1-(mesitylsulfonyl)-2-((*E*)-1-phenylethylidene)hydrazineyl)-2-phenylethylidene)-4-methylbenzenesulfonamide (**4f**). White solid, mp 181–183 °C. IR (KBr) *ν* 3062, 1600, 1551, 1354, 1304, 1141, 1088 cm^−1^; ^1^H NMR (400 MHz, DMSO-*d_6_*) *δ* 7.77 (d, *J* = 7.6 Hz, 2H), 7.60 (d, *J* = 7.2 Hz, 1H), 7.50 (t, *J* = 7.8 Hz, 2H), 7.34–7.17 (m, 7H), 7.03 (d, *J* = 7.2 Hz, 2H), 6.93 (s, 2H), 4.58 (s, 2H), 2.43 (s, 6H), 2.34 (s, 3H), 2.32 (s, 3H), 1.82 (s, 3H); ^13^C NMR (100 MHz, DMSO-*d_6_*) *δ* 181.3, 164.7, 143.9, 143.2, 140.3, 138.5, 135.0, 133.1, 132.4, 132.3, 132.0, 131.9, 129.4 (2C), 128.8, 128.7 (2C), 128.5 (2C), 127.9, 127.7 (2C), 127.1 (2C), 126.3 (2C), 21.8 (2C), 21.0 (2C), 20.7, 18.5; HRMS (ESI-TOF) (*m*/*z*). Calcd for C_32_H_33_N_3_O_4_S_2_, [M + H]^+^ 590.1985; found 590.1988.

4-methyl-*N*-((E)-1-(2-((*E*)-1-phenylethylidene)-1-tosylhydrazineyl)-2-(*p*-tolyl)ethylidene)benzenesulfonamide (**4g**). White solid, mp 159–160 °C. IR (KBr) *ν* 3062, 2920, 1596, 1566, 1367, 1174, 1142, 1085 cm^−1^; ^1^H NMR (400 MHz, DMSO-*d_6_*) *δ* 7.83 (d, *J* = 7.6 Hz, 2H), 7.61 (d, *J* = 8.0 Hz, 3H), 7.54–7.45 (m, 6H), 7.26 (d, *J* = 8.0 Hz, 2H), 7.02 (d, *J* = 7.6 Hz, 2H), 6.90 (d, *J* = 8.0 Hz, 2H), 4.19 (s, 2H), 2.47 (s, 3H), 2.41 (s, 3H), 2.26 (s, 3H), 1.74 (s, 3H); ^13^C NMR (100 MHz, DMSO-*d_6_*) *δ* 182.7, 165.4, 145.5, 143.5, 143.2, 138.6, 136.4, 135.1, 134.0, 132.3, 130.0, 129.6 (2C), 129.5 (2C), 129.1 (2C), 128.8 (2C), 128.5 (2C), 127.8 (2C), 126.5 (3C), 21.2 (2C), 20.7 (2C), 17.8; HRMS (ESI-TOF) (*m*/*z*). Calcd for C_31_H_31_N_3_O_4_S_2_, [M + H]^+^ 574.1829; found 574.1832.

4-methyl-*N*-((*E*)-1-(2-((*E*)-1-phenylethylidene)-1-tosylhydrazineyl)-2-(*m*-tolyl)ethylidene)benzenesulfonamide (**4h**). White solid, mp 146–148 °C. IR (KBr) *ν* 3062, 2920, 1598, 1569, 1489, 1359, 1367, 1294, 1142, 1087 cm^−1^; ^1^H NMR (400 MHz, DMSO-*d_6_*) *δ* 7.84 (d, *J* = 7.6 Hz, 2H), 7.62 (d, *J* = 8.0 Hz, 3H), 7.53–7.45 (m, 6H), 7.28 (d, *J* = 8.0 Hz, 2H), 7.11 (t, *J* = 7.6 Hz, 1H), 7.02 (d, *J* = 7.6 Hz, 1H), 6.87 (d, *J* = 7.6 Hz, 1H), 6.66 (s, 1H), 4.21 (s, 2H), 2.47 (s, 3H), 2.42 (s, 3H), 1.99 (s, 3H), 1.74 (s, 3H); ^13^C NMR (100 MHz, DMSO-*d_6_*) *δ* 182.6, 165.2, 145.6, 138.6, 137.7, 134.9, 134.0, 133.0, 132.4, 130.5 (2C), 129.6 (2C), 129.5 (2C), 128.7 (2C), 128.5 (2C), 127.8 (2C), 127.6, 126.5 (3C), 125.7, 21.2 (3C), 20.7, 17.6; HRMS (ESI-TOF) (*m*/*z*). Calcd for C_31_H_31_N_3_O_4_S_2_, [M + H]^+^ 574.1829; found574.1830.

*N*-((*E*)-2-(4-fluorophenyl)-1-(2-((*E*)-1-phenylethylidene)-1-tosylhydrazineyl) ethylidene)-4-methylbenzenesulfonamide (**4i**). White solid, mp 157–159 °C. IR (KBr) *ν* 3062, 1595, 1564, 1375, 1308, 1190, 1083 cm^−1^; ^1^H NMR (400 MHz, DMSO-*d_6_*) *δ* 7.85 (d, *J* = 8.0 Hz, 2H), 7.62 (d, *J* = 8.0 Hz, 3H), 7.54–7.45 (m, 6H), 7.27 (d, *J* = 8.0 Hz, 2H), 7.09–7.05 (m, 4H), 4.20 (s, 2H), 2.47 (s, 3H), 2.42 (s, 3H), 1.86 (s, 3H); ^13^C NMR (100 MHz, DMSO-*d_6_*) *δ* 182.5, 165.0, 161.2 (d, *J* = 256.7 Hz), 145.7, 143.6, 138.5, 135.0, 133.9, 132.4, 130.7 (2C), 129.7 (2C), 129.6 (2C), 129.2 (d, *J* = 3.1 Hz), 128.8 (2C), 128.5 (2C), 127.8 (2C), 126.5 (3C), 115.5 (d, *J* = 21.8 Hz), 21.1 (2C), 21.1 (d, *J* = 7.7 Hz), 17.9; HRMS (ESI-TOF) (*m*/*z*). Calcd for C_30_H_28_FN_3_O_4_S_2_, [M + H]^+^ 578.1578; found 578.1581.

*N*-((*E*)-2-(4-chlorophenyl)-1-(2-((*E*)-1-phenylethylidene)-1-tosylhydrazineyl) ethylidene)-4-methylbenzenesulfonamide (**4j**). White solid, mp 153–155 °C. IR (KBr) *ν* 3064, 1593, 1562, 1444, 1345, 1272, 1122, 1081 cm^−1^; ^1^H NMR (400 MHz, DMSO-*d_6_*) *δ* 7.84 (d, *J* = 8.4 Hz, 2H), 7.62 (d, *J* = 8.0 Hz, 3H), 7.53–7.45 (m, 6H), 7.30–7.27 (m, 4H), 7.02 (d, *J* = 8.8 Hz, 2H), 4.24 (s, 2H), 2.47 (s, 3H), 2.42 (s, 3H), 1.90 (s, 3H); ^13^C NMR (100 MHz, DMSO-*d_6_*) *δ* 182.5, 164.8, 145.7, 143.7, 138.4, 135.0, 133.8, 132.4, 132.1, 132.0, 130.6 (2C), 129.7 (2C), 129.6 (2C), 128.8 (2C), 128.5 (2C), 127.8 (2C), 126.5 (3C), 38.0, 21.1 (2C), 21.1, 18.0; HRMS (ESI-TOF) (*m*/*z*). Calcd for C_30_H_28_ClN_3_O_4_S_2_, [M + H]^+^ 594.1283; found 594.1285.

*N*-((*E*)-2-(4-bromophenyl)-1-(2-((*E*)-1-phenylethylidene)-1-tosylhydrazineyl) ethylidene)-4-methylbenzenesulfonamide (**4k**). White solid, mp 158–160 °C. IR (KBr) *ν* 3062, 1592, 1560, 1486 1369, 1282, 1142, 1082 cm^−1^; ^1^H NMR (400 MHz, DMSO-*d_6_*) *δ* 7.83 (d, *J* = 7.2 Hz, 2H), 7.62 (d, *J* = 8.0 Hz, 3H), 7.53–7.41 (m, 8H), 7.28 (d, *J* = 8.0 Hz, 2H), 6.95 (d, *J* = 8.0 Hz, 2H), 4.21 (s, 2H), 2.47 (s, 3H), 2.42 (s, 3H), 1.91 (s, 3H); ^13^C NMR (100 MHz, DMSO-*d_6_*) *δ* 182.4, 164.7, 145.7, 143.6, 138.4, 135.0, 133.8, 132.5, 132.4, 131.5, 132.0, 130.8 (2C), 129.7 (2C), 129.6 (2C), 128.8 (2C), 128.5 (2C), 127.8 (2C), 126.5 (3C), 120.3, 21.1 (2C), 18.0; HRMS (ESI-TOF) (*m*/*z*). Calcd for C_30_H_28_BrN_3_O_4_S_2_, [M + H]^+^ 638.0778; found 638.0779.

4-methyl-*N*-((*E*)-1-(2-((*E*)-1-phenylethylidene)-1-tosylhydrazineyl)octylidene) benzenesulfonamide (**4l**). White solid, mp 103–105 °C. IR (KBr) *ν* 3063, 2864, 1595, 1338, 1264, 1155, 1076 cm^−1^; ^1^H NMR (400 MHz, DMSO-*d_6_*) *δ* 8.00 (d, *J* = 7.2 Hz, 2H), 7.63 (t, *J* = 7.6 Hz, 1H), 7.57–7.53 (m, 6H), 7.40 (d, *J* = 8.4 Hz, 2H), 7.29 (d, *J* = 8.0 Hz, 2H), 2.75 (d, *J* = 7.6 Hz, 2H), 2.56 (s, 3H), 2.44 (s, 3H), 2.40 (s, 3H), 1.39 (s, 2H), 1.17–1.08 (m, 8H), 0.75 (t, *J* = 6.8Hz, 3H); ^13^C NMR (100 MHz, DMSO-*d_6_*) *δ* 181.5, 167.9, 145.6, 143.3, 138.9, 135.4, 134.1, 132.4, 129.7 (2C), 129.6 (2C), 128.9 (2C), 128.4 (2C), 127.8 (2C), 126.3 (2C), 32.5, 30.9, 28.8, 27.8, 24.9, 21.9, 21.3, 21.1, 18.7, 13.9; HRMS (ESI-TOF) (*m*/*z*). Calcd for C_30_H_37_N_3_O_4_S_2_, [M + H]^+^ 568.2298; found 568.2231.

4-methyl-*N*-((*E*)-1-(2-((*E*)-1-phenylethylidene)-1-tosylhydrazineyl)-2-(thiophen-2-yl)ethylidene)benzenesulfonamide (**4m**). Yellow solid, mp 67–69 °C. IR (KBr) *ν* 3062, 2927, 2866, 1590, 1369, 1307, 1153, 1087 cm^−1^; ^1^H NMR (400 MHz, DMSO-*d_6_*) *δ* 7.85 (t, *J* = 6.8 Hz, 4H), 7.65 (d, *J* = 9.2 Hz, 3H), 7.48 (d, *J* = 7.8 Hz, 2H), 7.36 (d, *J* = 7.8 Hz, 2H), 7.11 (d, *J* = 7.8 Hz, 3H), 6.86–6.82 (m, 2H), 4.58 (s, 2H), 2.50 (s, 3H), 2.41 (s, 3H), 2.00 (s, 3H); ^13^C NMR (100 MHz, DMSO-*d_6_*) *δ* 183.3, 163.9, 145.4, 143.4, 139.2, 135.8, 134.4, 134.2, 132.2, 129.4 (2C), 129.3 (2C), 129.2 (2C), 128.8 (2C), 128.1, 127.9 (2C), 127.1 (2C), 127.0, 125.4, 33.8, 21.9, 21.8, 18.2; HRMS (ESI-TOF) (*m*/*z*). Calcd for C_28_H_27_N_3_O_4_S_3_, [M + H]^+^ 565.1237; found 565.1239.

*N*-(2-phenyl-1-(2-((*E*)-1-phenylethylidene)-1-tosylhydrazineyl)ethylidene) benzenesulfonamide (**4n**). White solid, mp 149–151 °C. IR (KBr) *ν* 3062, 1589, 1561, 1494, 1365, 1282, 1140, 1085 cm^−1^; ^1^H NMR (400 MHz, DMSO-*d_6_*) *δ* 7.82 (d, *J* = 8.0 Hz, 2H), 7.76 (d, *J* = 6.8 Hz, 3H), 7.70–7.60 (m, 3H), 7.53–7.46 (m, 4H), 7.27–7.20 (m, 5H), 7.02 (d, *J* = 6.8 Hz, 2H), 4.23 (s, 2H), 2.41 (s, 3H), 1.74 (s, 3H); ^13^C NMR (100 MHz, DMSO-*d_6_*) *δ* 182.7, 165.7, 145.6, 141.3, 135.0, 133.9, 133.1, 133.0, 132.4, 129.6, 129.3 (2C), 128.9 (2C), 128.8 (2C), 128.6 (2C), 128.5 (2C), 127.8 (2C), 127.2, 126.4 (3C), 21.2 (2C), 17.7; HRMS (ESI-TOF) (*m*/*z*). Calcd for C_29_H_27_N_3_O_4_S_2_, [M + H]^+^ 546.1516; found 546.1519.

4-chloro-*N*-(2-phenyl-1-(2-((*E*)-1-phenylethylidene)-1-tosylhydrazineyl) ethylidene)benzenesulfonamide (**4o**). White solid, mp 141–143 °C. IR (KBr) *ν* 3067, 1592, 1554, 1493, 1341, 1308, 1146, 1081 cm^−1^; ^1^H NMR (400 MHz, DMSO-*d_6_*) *δ* 7.83 (d, *J* = 8.0 Hz, 2H), 7.75 (t, *J* = 9.6 Hz, 4H), 7.62 (t, *J* = 7.6 Hz, 1H), 7.51 (t, *J* = 8.0 Hz, 4H), 7.29–7.20 (m, 5H), 7.02 (t, *J* = 6.8 Hz, 2H), 4.15 (s, 2H), 2.42 (s, 3H), 1.77 (s, 3H); ^13^C NMR (100 MHz, DMSO-*d_6_*) *δ* 182.8, 165.5, 145.7, 140.2, 138.0, 135.0, 134.0, 133.0, 132.4, 129.6 (2C), 129.4 (3C), 128.8, 128.7 (2C), 128.6 (2C), 128.4 (2C), 128.3 (2C), 127.8 (2C), 127.2, 21.2 (2C), 17.8; HRMS (ESI-TOF) (*m*/*z*). Calcd for C_29_H_26_ClN_3_O_4_S_2_, [M + H]^+^ 580.1126; found 580.1128.

4-bromo-*N*-(2-phenyl-1-(2-((*E*)-1-phenylethylidene)-1-tosylhydrazineyl) ethylidene)benzenesulfonamide (**4p**). White solid, mp 139–140 °C. IR (KBr) *ν* 3066, 1594, 1554, 1493, 1374, 1309, 1145, 1083 cm^−1^; ^1^H NMR (400 MHz, DMSO-*d_6_*) *δ* 7.89 (d, *J* = 8.4 Hz, 2H), 7.83 (t, *J* = 7.6 Hz, 2H), 7.68 (t, *J* = 7.6 Hz, 2H), 7.62 (t, *J* = 7.2 Hz, 1H), 7.53–7.49 (m, 4H), 7.29–7.20 (m, 5H), 7.01 (t, *J* = 7.2 Hz, 2H), 4.23 (s, 2H), 2.42 (s, 3H), 1.76 (s, 3H); ^13^C NMR (100 MHz, DMSO-*d_6_*) *δ* 182.8, 165.5, 145.7, 140.6, 135.0, 134.0, 133.0, 132.4 (3C), 129.6 (2C), 128.8 (4C), 128.7 (2C), 128.4 (3C), 128.3 (2C), 127.2 (2C), 127.0, 21.2 (2C), 17.8; HRMS (ESI-TOF) (*m*/*z*). Calcd for C_29_H_26_BrN_3_O_4_S_2_, [M + H]^+^ 624.0621; found 624.0622.

4-methoxy-*N*-(2-phenyl-1-(2-((*E*)-1-phenylethylidene)-1-tosylhydrazineyl) ethylidene)benzenesulfonamide (**4q**). White solid, mp 143–145 °C. IR (KBr) *ν* 3010, 1592, 1561, 1492, 1367, 1296, 1144, 1082 cm^−1^; ^1^H NMR (400 MHz, DMSO-*d_6_*) *δ* 7.82 (d, *J* = 7.6 Hz, 2H), 7.69 (t, *J* = 8.4 Hz, 2H), 7.62 (t, *J* = 7.2 Hz, 1H), 7.51 (t, *J* = 8.0 Hz, 4H), 7.29 (d, *J* = 8.0 Hz, 2H), 7.24–7.17 (m, 5H), 7.01 (d, *J* = 6.8 Hz, 2H), 4.24 (s, 2H), 3.92 (s, 3H), 2.42 (s, 3H), 1.73 (s, 3H); ^13^C NMR (100 MHz, DMSO-*d_6_*) *δ* 183.0, 165.4, 163.1, 146.0, 135.5, 134.4, 133.6, 132.7, 130.0, 129.3 (2C), 129.2 (3C), 129.1 (4C), 129.0 (2C), 128.9 (2C), 128.2 (2C), 127.5, 114.8, 56.3, 21.7 (2C), 18.1; HRMS (ESI-TOF) (*m*/*z*). Calcd for C_30_H_29_N_3_O_5_S_2_, [M + H]^+^ 576.1622; found 576.1621.

1-phenyl-*N*-(2-phenyl-1-(2-((*E*)-1-phenylethylidene)-1-tosylhydrazineyl) ethylidene)methanesulfonamide (**4r**). White solid, mp 125–127 °C. IR (KBr) *ν* 3063, 2972, 1590, 1576, 1493, 1365, 1293, 1173, 1086 cm^−1^; ^1^H NMR (400 MHz, DMSO-*d_6_*) *δ* 7.85–7.79 (m, 4H), 7.62 (t, *J* = 7.2 Hz, 1H), 7.56–7.50 (m, 4H), 7.21 (t, *J* = 6.8 Hz, 3H), 7.01 (d, *J* = 7.2 Hz, 2H), 4.18 (s, 2H), 3.04(t, *J* = 7.6 Hz, 2H), 2.46 (s, 3H), 1.75 (s, 3H), 1.69 (s, 2H), 1.02 (t, *J* = 7.6 Hz, 3H); ^13^C NMR (100 MHz, DMSO-*d_6_*) *δ* 182.5, 165.5, 145.7, 135.1, 134.6, 133.1, 132.3, 129.9 (3C), 128.9 (2C), 128.8 (3C), 128.6 (2C), 128.3 (3C), 127.8 (3C), 127.1, 56.0 (2C), 21.2, 17.6, 16.8, 12.6; HRMS (ESI-TOF) (*m*/*z*). Calcd for C_30_H_29_N_3_O_4_S_2_, [M + H]^+^ 560.1672; found 560.1676.

## 4. Conclusions

We developed an effective copper-catalyzed three-component one-pot synthesis of *N*-sulfonyl amidines from terminal alkynes, sulfonyl azides and weak nucleophilic sulfonyl hydrazine. The synthetic pathway extended the applications of the CuAAC/ring-opening reaction, and we expect that this methodology and *N*-sulfonyl amidines products could be applied to organic synthesis.

## Data Availability

The data is contained within the article.

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
