# Peer review of "Copper-Catalyzed One-Pot Synthesis of *N*-Sulfonyl Amidines from Sulfonyl Hydrazine, Terminal Alkynes and Sulfonyl Azides"

_molecules, 2021, doi:10.3390/molecules26123700_

Round 1

Reviewer 1 Report

In this manuscript, the authors investigated the one-pot synthesis of N-sulfonyl amidines via Cu-catalyzed three-component conjugate reaction. Although, this study is relatively basic, it opens new avenues to the synthesis of new amidine derivatives. I therefore recommend acceptance in its current form.

Author Response

Thanks for the Reviewer’s recognition and we will continue to work hard to achieve better achievements.

Besides, the full text was checked carefully and some words and grammars were corrected.

Reviewer 2 Report

In the paper entitled "Copper-catalyzed one-pot synthesis of N-sulfonyl amidines from sulfonyl hydrazine, terminal alkynes, and sulfonyl azides" the authors present an easy one-pot synthesis of sulfonyl-amidines, starting from sulfonyl hydrazine, various terminal alkynes, and sulfonyl azides, using Copper (I) iodide and triethylamine as catalyst and benzene as reaction media. Overall, the paper is well structured and written, and the compounds obtained are well characterised. I suggest acceptance for publication, after some minor changes:

  1. The characterisation of all compounds should be included in the main article;
  2. All spectra (MS and IR included) should be provided;
  3. Minor English language editing.

Author Response

1. The characterisation of all compounds should be included in the main article;

Reply: We accepted the suggestion of reviewer 2. We have moved the experiment section and characterisation of all compounds to the main text in the revised manuscript and reorganize the SI document.

2. All spectra (MS and IR included) should be provided;

Reply: We are sorry that compounds 4j and 4l’s date are omitted, and thanks for reminding me. We have provided the relevant datas in the revised manuscript.

3. Minor English language editing.

Reply: We accepted the suggestion of reviewer 2 and the full text was checked carefully and some words and grammars were corrected.

Reviewer 3 Report

This submission by Yang and coworkers describes the Cu-catalyzed three-component reaction for the synthesis of N-sulfonyl amidine. N-sulfonyl amidines are important motifs found in many biologically relevant compounds. The synthesis of N-sulfonyl amidines begins with sulfonyl hydrazine, terminal alkynes, sulfonyl azides, and this multi-component reaction is facilitated by a Cu-catalyst. The authors optimized the reaction by varying several parameters and found that a combination of CuI/TEA/Tol. best for this response. Next, the authors examine the substrate scope of the reaction and as listed in Figure 2, the reaction behaves well, with a range of electronically diverse, hydrazine and alkynes. Transformation studies in detail and good experimental efforts and complete analytical data are presented to elucidate the synthesized compound. Hence in my opinion this manuscript meets the requirements of the Molecules, and it would be accepted after minor revisions.

Comments on minor revision:

Authors have widely explored phenyl carbocycles core as a substituent on the hydrazine and alkyne, I would recommend Author should examine a few heterocyclic substituents (pyridine, thiophane, and furan) either on hydrazine or on alkynes, and the result should be discussed in the revised manuscript.

Author Response

Reply: Thanks for the reviewer’s recognition and we accepted the suggestion. We add the 2-thienyl substituent product 4m to scheme 2 and add the discussion in the revised manuscript page 3 line 75. Also, we rearrange all product numbers in the text and supporting information file.